# Patients with Severe Trauma Having an Injury Severity Score of 24 and above Develop Nutritional Disorders

**DOI:** 10.3390/diagnostics14121307

**Published:** 2024-06-20

**Authors:** Minori Mizuochi, Junko Yamaguchi, Nobutaka Chiba, Kosaku Kinoshita

**Affiliations:** Division of Emergency and Critical Care Medicine, Department of Acute Medicine, Nihon University School of Medicine, Tokyo 173-8610, Japan; orita.minori@nihon-u.ac.jp (M.M.); chiba.nobutaka@nihon-u.ac.jp (N.C.); kinoshita.kosaku@nihon-u.ac.jp (K.K.)

**Keywords:** severe injury, controlling nutritional status score, injury severity score, malnutrition

## Abstract

In this single-center, retrospective, observational study, we aimed to assess the severity at which patients with trauma tend to develop metabolic disturbances that worsen their Controlling Nutritional Status (CONUT) scores. Participants were general adult patients with trauma hospitalized for at least one week. Injury Severity Scores (ISSs) at admission and CONUT scores one week later were calculated, and correlation coefficients were examined. The receiver operating characteristic (ROC) curve was used to calculate the ISS cutoff value for a CONUT score of 5 or more on day 7 of hospitalization. The ISS was assessed using multiple logistic regression analysis to determine whether it predicts worse nutritional status. Forty-nine patients were included. ISSs correlated with CONUT scores on day 7 (r = 0.373, *p* = 0.008). Using the ROC curve, the cutoff value for the ISS was 23.5. Multiple logistic regression analyses showed that a high ISS (odds ratio [OR], 1.158; 95% confidence interval [CI], 1.034–1.296; *p* = 0.011) and older age (OR, 1.094; 95% CI, 1.027–1.165; *p* = 0.005) were associated with a CONUT score 5 or more on day 7 of hospitalization. Patients with trauma with an ISS of 24 or higher have worsening CONUT scores during hospitalization; these patients require careful nutritional management.

## 1. Introduction

Malnutrition in patients that are critically ill, admitted to the intensive care unit (ICU) is associated with poor clinical outcomes [1], and patients with trauma are no exception to this trend [2,3,4]. Patients with multiple traumas are in a hypercatabolic state and are nutritionally impaired, as was originally proposed by Cuthbertson in the 1940s [5,6]. Such patients are more likely to be malnourished during their hospitalization. Therefore, it is important to perform nutritional screening in patients with severe trauma for appropriate intervention. However, the relationship between the severity of trauma and subsequent malnutrition remains unclear. The Controlling Nutritional Status (CONUT) [7,8] score is a method for nutritional assessment that measures serum albumin, total cholesterol, and lymphocyte count. The CONUT score was chosen as the assessment tool because the assessment methods that are considered the gold standard for nutritional assessment, such as the Subjective Global Assessment (SGA) and the Nutritional Risk Screening (NRS) 2002, are complicated and require detailed history taking and muscle strength assessment, making them difficult to use. Meanwhile, the CONUT score is easy to use in emergency settings, and changes over time can be assessed after admission. Recent reports have shown an association between CONUT scores and prognosis and complications [9,10,11,12,13,14]. With regard to trauma, studies have reported relationships between the CONUT score, which indicates nutritional status, and the prognosis of patients with head trauma [15,16] and cervical cord injury [17]. However, no previous studies have examined the association between worsening CONUT scores and the severity of systemic trauma. In this study, we aimed to assess the nutritional status of patients with acute severe trauma using the CONUT score and clarify its association with patient outcomes.

### 1.1. Injury Severity Score (ISS)

The ISS is an established medical scoring system used to evaluate the severity of trauma injury [18,19]. The scores correlate with mortality, morbidity, and length of hospital stay after trauma. Major trauma (or polytrauma) is defined as an ISS of ≥15 [19].

### 1.2. CONUT Score

The CONUT score is a measure for nutritional assessment proposed by de Ulíbarri et al. of Spain in 2003, which evaluates nutritional status by measuring serum levels of albumin, total cholesterol, and lymphocyte counts. These items reflect protein metabolism, lipid metabolism, and immunocompetence, respectively. A score of 0–1 is defined as normal; 2–4, as mildly abnormal; 5–8, as moderately abnormal; and 9–12, as severely abnormal. A score of 5 or higher is proposed as the standard for intervention. Since this scoring system utilizes only relatively routine items, it is considered a simple, objective, and easy system that can be evaluated at various facilities. It has been reported to correspond with the SGA, the gold standard for nutritional assessment [7,8]. It has also been reported to have high concordance with the Global Leadership Initiative on Malnutrition score, which can assess nutritional status in acute-inflammation-related diseases [20].

## 2. Materials and Methods

### 2.1. Study Participants

This single-institution, retrospective, observational study used the database of patients treated for trauma at our hospital and was approved by the Clinical Research Review Committee of Nihon University Hospital (RK-20220901). The requirement for informed consent was waived by the approving authorities owing to the retrospective nature of this study.

Patients enrolled in this study included those aged ≥18 years who were admitted to the ICU of the hospital between May 2018 and April 2020. These patients were assessed for their ISSs [18,19]. All data for this study were obtained from hospital databases and patients’ clinical records. Day 7 of hospitalization was established as a reasonable period to observe changes in nutritional status because the ESPEN guidelines [21] set a standard of 7 days for the acute phase in metabolic changes after invasion. The included patients had already been diagnosed with trauma and received treatment for 1 week or more at our hospital. Patients who had received treatment at other hospitals before admission and patients with a CONUT score of 5 or higher at admission were excluded from this study. Physical examination and whole-body computed tomography were performed at the site of trauma to confirm the diagnosis. Peripheral whole-blood samples were collected from the patients upon admission and 1 week later. Patient information and laboratory data, including age, sex, Sequential Organ Failure Assessment (SOFA) [22,23], and Acute Physiology and Chronic Health Evaluation (APACHE) II scores [24], were recorded. The body mass index (BMI) and CONUT scores [7] of the patients were recorded upon admission to assess their nutritional status at baseline and 1 week after admission. The total calorie intake (oral, enteral, and intravenous nutrition) for each patient over 7 days was calculated to assess nutritional support during hospitalization. We typically administer an energy dose of 25 kcal/standard body weight/day, according to the Japanese nutrition guidelines for critical care patients and several international guidelines [1,21,25]. The ISS was calculated upon admission to assess the severity of trauma.

Patients were classified into two groups according to the CONUT score measured on day 7: the high CONUT score group (CONUT score ≥ 5) and the low CONUT score group (CONUT score ≤ 4). Injury site, injury severity, and prognosis were compared between the two groups.

### 2.2. Statistical Analyses

All statistical analyses were performed using SPSS version 28 (IBM Corp., Armonk, NY, USA) and JMP ver. 14.2 (SAS Institute, Cary, NC, USA). Data are presented as mean values (standard deviation [SD]), median values (interquartile range), or number of cases (%). Statistical significance was set at *p* < 0.05. Continuous variables were compared using Student’s *t*-test or the Mann–Whitney U test and Wilcoxon’s signed-rank test, as appropriate. Chi-square or Fisher’s exact probability tests were performed for categorical variables. The correlation between the ISS and CONUT scores was examined using Spearman’s rank correlation coefficient. Using receiver operating characteristic (ROC) curve analysis, we determined the optimal cutoff points for the ISS to have a CONUT score of at least 5 points at a significance level of 5%. Multiple logistic regression analysis was used to predict whether the CONUT score would be 5 or higher. Variables with *p*-values of <0.05 in the bivariate models (age, SOFA score, operation, ISS) were transferred to the multivariate models. APACHEII was excluded because it includes age as part of the scoring item.

## 3. Results

A total of 216 patients with trauma were initially enrolled in this study. After excluding patients who had commenced treatment at another hospital, those with incomplete data, those aged <18 years, and patients with a CONUT score of 5 or higher upon admission, the final study cohort consisted of 49 patients with trauma (42 men and 9 women), as depicted in Figure 1 and Table 1. The sites of trauma were as follows: head (*n* = 21), spine (*n* = 16), chest (*n* = 10), abdomen (*n* = 5), and extremities (*n* = 21). The patients had a median age of 52.0 years (quartile range was 37.5–67.5 years) and a median BMI of 24.2 kg/m^2^ (quartile range was 20.3–25.6 kg/m^2^). None of these 49 patients died. A total of 25 patients were categorized into the low CONUT score group, and 24 into the high CONUT score group on day 7 of hospitalization. There was a significant deterioration in CONUT scores by day 7 after admission compared with the scores on the day of admission (Figure 2; median 1 versus 4, *p* < 0.001).

Albumin levels, lymphocyte count, and total cholesterol level on day 7 were significantly lower in the high CONUT score group than in the low CONUT score group (*p* < 0.001, *p* < 0.001, and *p* < 0.001, respectively). When the patients were divided into the high and low CONUT score groups on day 7, the high CONUT score group had a longer ICU stay (median days: 8.0 vs. 11.5, *p* = 0.029) and total length of stay (median days: 25.0 vs. 44.0, *p* = 0.018) compared to the low CONUT score group (Figure 3).

No correlation was found between the ISS and CONUT scores at admission (r = 0.209, *p* = 0.150); however, the ISSs at admission were correlated with CONUT scores on day 7 (r = 0.373, *p* = 0.008; Figure 4). To assess the severity of trauma as a risk factor for malnutrition, the diagnostic performance of the ISS (sensitivity, specificity, and negative and positive predictive values) was calculated, an ROC curve was constructed to calculate the cutoff value of the ISS for a CONUT score of ≥5 on day 7, and the corresponding area under the ROC curve (AUROC) was calculated (Figure 5). The ROC curve analysis showed that the baseline ISS that distinguished between the low CONUT score (CONUT score ≤ 4) and the high CONUT score groups of patients (CONUT score ≥ 5) on day 7 was 23.5, which maximized the Youden Index with a sensitivity and specificity of 0.63 and 0.80, respectively. The AUROC values were 0.69 (95% confidence interval [CI] 0.54–0.84). Variables with *p*-values of <0.05 in the bivariate models were age, SOFA score, operation, ISS, and APACHE II (Table 2). APACHE II was excluded because it includes age as part of the scoring item. Multiple logistic regression analyses showed that high ISSs (odds ratio [OR], 1.158; 95% CI, 1.034–1.296; *p* = 0.011) and older age (OR, 1.094; 95% CI, 1.027–1.165; *p* = 0.005) were associated with a CONUT score of ≥5 on day 7 of hospitalization. SOFA scores and operation were not correlated with a CONUT score of ≥5 on day 7 of hospitalization (Table 3). The characteristics of each trauma site were also examined (Appendix A), and we analyzed the abdominal cases because they may be more severe than other trauma cases (Appendix A). The median hospital stay for the abdominal cases was 30 days (quartile range, 24.5–56.0), with a median ICU stay of 15 days (interquartile range, 12.0–26.5). The median CONUT scores of the five patients with abdominal trauma at admission and day 7 were 2 and 5, respectively. The ISS for patients with abdominal trauma was a median of 36 points (interquartile range, 12.5–37.5). In contrast, the median hospital stay for patients with non-abdominal trauma was 32 days (interquartile range 19.3–49.5), and the median ICU stay was 9 days (interquartile range 5.3–12.0). The median CONUT scores for patients with non-abdominal trauma were 1 and 4 points at admission and on day 7, respectively. The ISS for patients with non-abdominal trauma was a median of 17 (interquartile range; 9–25). Energy intake for 7 days (kcal) was a median of 6234.0 for patients with abdominal trauma and 6239.5 for patients with non-abdominal trauma.

In comparing the groups of patients with abdominal trauma and with non-abdominal trauma, the patients with abdominal trauma had a significantly longer ICU stay (15.0 days vs. 9.0 days, *p* = 0.005). However, there were no significant differences between the groups in terms of hospital stay, CONUT score at admission, CONUT score on day 7, energy intake over 7 days, and ISS (*p* = 0.596, 1.0, 0.712, 0.828, and 0.146, respectively). The clinical course of trauma patients with and without surgery was significantly different. The clinical course of trauma patients with surgical treatment was significantly lower in terms of energy intake (kcal) over 7 days (*p* = 0.029) than that of trauma patients without surgical treatment, and CONUT scores on day 7 were higher (*p* = 0.004). However, there was no significant difference in sufficiency rate (*p* = 0.206) (Appendix A).

## 4. Discussion

In this study, we examined the correlation between the ISS, which indicates the severity of the trauma, and the CONUT score, which indicates the nutritional status. The results of the present study revealed a correlation between the ISS and CONUT scores at admission (r = 0.373, *p* = 0.008). Multivariate analysis showed the ISS as a predictor of a CONUT score of ≥5 on day 7. When the ROC curve was examined based on whether the CONUT score on day 7 was ≥5, the ISS cutoff value was 23.5. Patients with trauma vary in the severity of their condition, and not all of them have metabolic disorders or experience deterioration of nutritional status. The present study suggests that worsening CONUT scores in patients with trauma with an ISS of ≥24 should be noted. Although the multivariate analysis showed that surgical therapy was not an independent predictor of a worse CONUT score (Table 3), a comparison of the two groups, those who received surgical therapy and those who did not receive surgical therapy, showed a lower ISS cutoff value in the group that received surgical therapy. The ISS cutoff value was 21.5 in the surgery group (the area under the ROC (AUROC) value was 0.71, and the 95% confidence interval (CI) was 0.51–0.90). In contrast, The ISS cutoff value was 27.5 in non-surgical groups (the area under the ROC (AUROC) value was 0.60, and the 95% confidence interval (CI) was 0.29–0.90) (Appendix A). The CONUT score measures albumin, total lymphocyte count, and total cholesterol levels in the blood. There are many reports on albumin in relation to prognosis in critically ill patients, and the same applies to trauma [26,27,28,29,30,31]. For total cholesterol, it has been reported that it predicts increased length of hospital stays in patients with trauma [32] and that hypocholesterolemia is associated with mortality in patients with head trauma [33]. Although there are few reports on lymphocyte counts in general patients with trauma, in recent years, several reports have suggested that the neutrophil/lymphocyte ratio is an indicator of prognosis in patients with head injury [34]. In brain injury following ischemic stroke, lymphocytes are said to contribute to the repair of the brain tissue [35], and there are also reports of decreased lymphocyte counts in patients with head trauma [36].

Previously, de Ulíbarri et al. defined nutritional impairment as a CONUT score of ≥5 [7]. In this study, patients with a CONUT score of ≤4 at admission were included. However, the median CONUT score worsened on day 7, which was thought to be affected by metabolic disorders due to invasive stress from trauma, with 24 of the 49 patients having a CONUT score of ≥5. The metabolic state that affects the nutritional status after traumatic injury is thought to change over time, as in the ebb and flow phases previously described by Cuthbertson [37]. Cuthbertson stated that the ebb phase, in which metabolism is suppressed, lasts approximately 3 days after injury, and the flow phase, in which metabolism increases, lasts 1 to 3 weeks. In a subsequent study, Uehara et al. [38] found that patients with severe trauma had increased resting energy expenditure (REE) during the second week of hospitalization. Monk et al. [39] found that patients with severe trauma had increased REE on the third day after injury, with peak REE on the 10th day and increased REE for at least 24 days. Vasileiou et al. [40] reported an increase in REE in patients with trauma observed on day 7 after admission to the ICU, which persisted until day 14.

The guidelines issued by the American Society for Parenteral and Enteral Nutrition and Society of Critical Care Medicine call for caution in cases of multiple or severe trauma because of the persistence of the catabolic phase up to the first week of hospitalization [1]. In this previous study, nutritional assessment was performed on the day of admission and after the first week, and patients with severe trauma had a deteriorated nutritional status by the first week. This recommendation of that study corresponds with those in our study.

In the present study, we did not examine the association with mortality because no patient died. However, the group with a CONUT score of ≥5 showed prolonged ICU stay and hospitalization. These results suggest that trauma patients with an ISS of ≥24 are more likely to be malnourished with a CONUT score of ≥5 points during the course of hospitalization. Thus, care should be taken to ensure appropriate nutritional management for such patients, for example, by referring to various nutritional management guidelines for patients who are critically ill.

Our study has some limitations. First, this study was performed retrospectively at a single center, and a high proportion of patients had head trauma only. Moreover, mortality was not studied because no patient died.

Target energy requirements for the acute 7-day period were not always met, despite nutritional therapy being provided with estimated amounts (Appendix A).

Energy requirements are expected to vary in each trauma patient based on trauma severity, surgical invasion, and other factors. However, our institution could not provide nutritional management using indirect calorimetry to calculate the specific energy requirements of each trauma patient.

Although the BMI of the trauma patients in this study was standardized to 20.3–25.6 kg/m^2^, allowing for the use of estimation formulas, the results showed that the energy intake over 7 days ranged from 4764 to 7321 kcal per week. The median sufficiency rate of energy intake during these 7 days was only 49% (Table 1).

Early enteral nutrition in our hospital is initiated within 48 h to reduce the complication rate of infections, including in trauma cases where energy could not be provided on the first day due to surgical treatment of the abdomen or other reasons. As per the ESPEN guideline [21], small doses should be started and gradually titrated up to the goal within 1 week. As a result, the energy intake was considered inadequate.

Further studies with a larger number of cases that monitor metabolic dynamics are needed to validate the findings of this study.

## 5. Conclusions

Patients with trauma with an ISS of ≥24 on day 7 of admission may become more malnourished and require careful nutritional management upon admission to improve their outcomes.

## Figures and Tables

**Figure 1 diagnostics-14-01307-f001:**
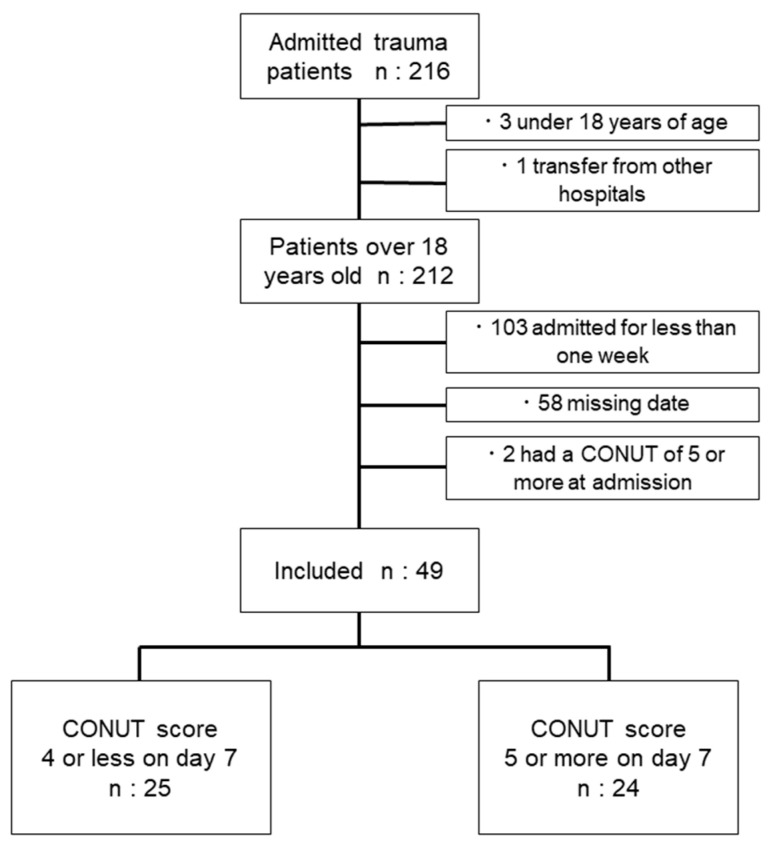
Flowchart of patients included in the study and their classification based on whether the CONUT score was 5 or higher on day 7. Patients were divided into groups of 25:24.

**Figure 2 diagnostics-14-01307-f002:**
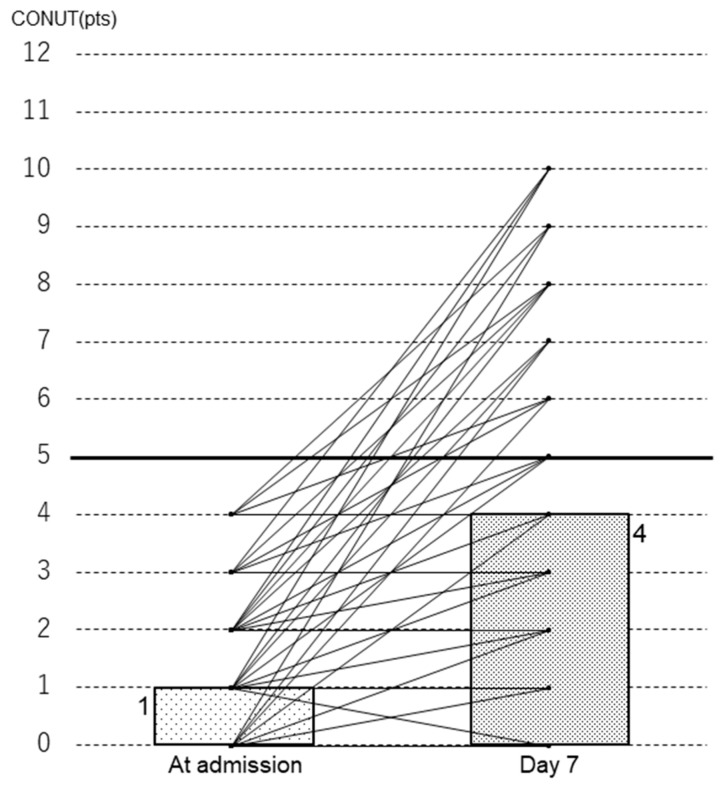
CONUT scores at admission and on day 7 of admission of the 49 patients. The thick horizontal line indicates a CONUT score of 5. The upper ends of the boxes are the median values for each group. The line connecting the score at admission and the score on day 7 illustrates the progression of each patient’s score. CONUT score, Controlling Nutritional Status score.

**Figure 3 diagnostics-14-01307-f003:**
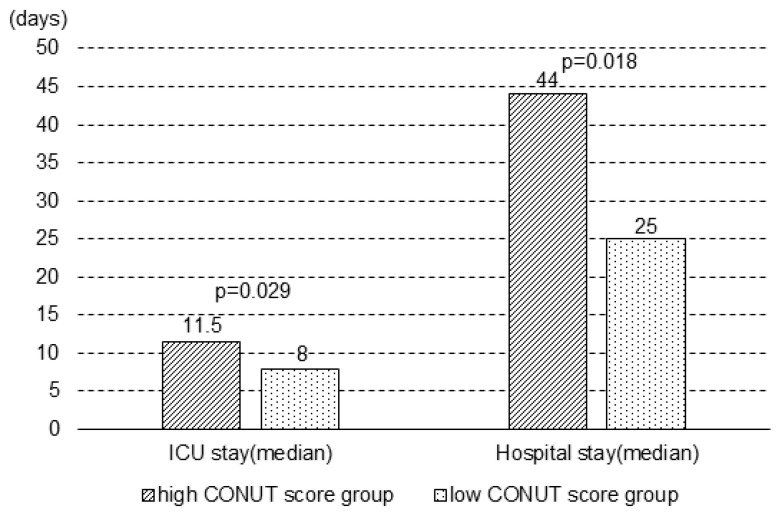
Duration of ICU stay and hospital stay between the two groups on day 7. The upper ends of the boxes are the median values for each group. CONUT score, Controlling Nutritional Status score; ICU, intensive care unit.

**Figure 4 diagnostics-14-01307-f004:**
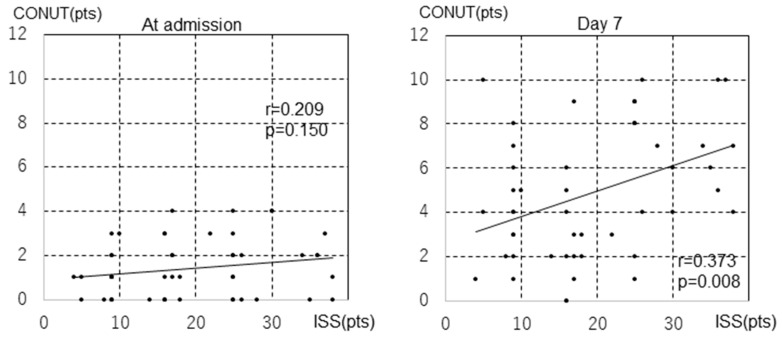
Correlation between the CONUT score and ISS at admission and on day 7 of admission. The inclined line is the regression line. CONUT score, Controlling Nutritional Status score; ISS, Injury Severity Score.

**Figure 5 diagnostics-14-01307-f005:**
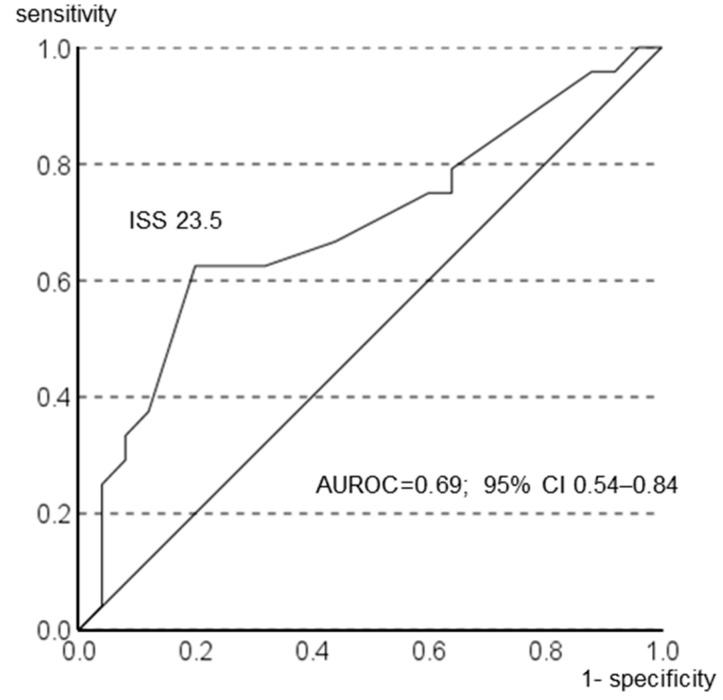
Receiver operating characteristic (ROC) curves for the Injury Severity Score (ISS) to assess correlation with high Controlling Nutritional Status scores on day 7. The area under the ROC (AUROC) value was 0.69, and the 95% confidence interval (CI) was 0.54–0.84. The ISS cutoff value was 23.5.

**Table 1 diagnostics-14-01307-t001:** Characteristics of patients at admission (*n* = 49).

	Median (Quartile Range)
Age (years)	52.0 (37.5–67.5)
Sex (M:F)	40:9
BMI	24.1 (20.3–25.6)
Energy intake during 7 days (kcal)	6234.0 (4764.0–7321.0)
Sufficiency rate * (%)	49.3 (37.9–66.9)
Site of injury	
Head	21
Spine	16
Chest	10
Abdomen	5
Extremities	21
Number of surgeries [*n*/ALL (%)]	29 [59.2]
APACHE II score	9.0 (5.0–14.0)
SOFA score	3.0 (2.0–4.0)
ISS score	17.0 (9.0–25.5)
Normal category (CONUT0-1) **	16.0 (9.0–25.0)
Light category (CONUT2-4)	5.0 (16.0–30.0)
ICU stay (day)	10.0 (6.0–13.0)
Hospital stay (day)	32.0 (20.5–49.0)
Transthyretin (mg/dL)	27.7 (24.3–29.8)
Phosphorus (mg/dL)	3.1 (2.6–3.7)
Magnesium (mg/dL)	1.9 (1.8–2.2)
Zinc (μg/dL)	68.0 (57.0–79.0)
CONUT score at admission	1.0 (0.0–2.0)
Normal category	28
Light category	21
Moderate category	0
Severe category	0
Lymphocyte count (/μL)	1883.7 (1149.0–2456.9)
T-cho (mg/dL)	189.0 (161.0–212.0)
Albumin (g/dL)	4.2 (3.9–4.5)

* Sufficiency rate = energy intake during 7 days/25 kcal × standard body weight × 7 days. Abbreviations: M, male; F, female; APACHE II, Acute Physiology and Chronic Health Evaluation II; SOFA, Sequential Organ Failure Assessment; ISS, Injury Severity Score; CONUT score, Controlling Nutritional Status score; BMI, body mass index. High CONUT score group: CONUT score ≥ 5. Low CONUT score group: CONUT score ≤ 4. ** CONUT0-1 corresponds to the undernutrition degree in the normal category, CONUT 2-4 corresponds to the undernutrition degree in the light category, CONUT 5-8 corresponds to the undernutrition degree in the moderate category, and CONUT 9-12 corresponds to the undernutrition degree in the severe category [7].

**Table 2 diagnostics-14-01307-t002:** Characteristics of the patients on day 7 (*n* = 49).

	High CONUT Score Group (*n* = 24)	Low CONUT Score Group (*n* = 25)	*p*-Value *
Age (years)	60.0 (43.0–77.8)	49.0 (28.5–64.5)	0.023
Sex (M/F)	18:6	22:3	0.289
BMI	22.2 (20.6–26.7)	24.4 (20.0–25.5)	0.741
Site of injury			
Head	12	9	0.393
Spine	5	11	0.128
Chest	5	5	1.000
Abdomen	3	2	0.667
Extremities	12	9	0.393
Number of surgeries [*n*/ALL (%)]	18 [36.7]	11 [22.4]	0.042
Energy intake during 7 days (kcal)	5572.0 (3928.0–6808.5)	6266.0 (4977.0–9339.4)	0.072
APACHE II score	12.0 (7.0–15.0)	5.0 (2.0–9.0)	<0.001
SOFA score	3.0 (2.0–4.0)	0.5 (0.0–2.75)	0.007
ISS	25.0 (11.5–33.0)	16.0 (9.0–20.0)	0.022
Transthyretin (mg/dL)	28.3 (24.2–30.2)	26.0 (23.8–29.4)	0.529
Phosphorus (mg/dL)	3.2 (2.6–4.5)	3.1 (2.6–3.4)	0.332
Magnesium (mg/dL)	2.0 (1.8–2.4)	1.9 (1.8–2.1)	0.739
Zinc (μg/dL)	67.5 (54.0–79.5)	73.0 (62.0–79.0)	0.638
CONUT score at admission	2.0 (1.0–3.0)	1.0 (0.0–1.5)	0.008
Normal category	9	19	0.010
Light category	15	6	0.010
Lymphocyte count (/μL)	1581.1 (1000.7–2430.9)	2157.4 (1451.0–2780.6)	0.129
T-cho (mg/dL)	182.0 (151.8–205.3)	198.0 (162.5–214.0)	0.412
Albumin (g/dL)	4.1 (3.7–4.3)	4.3 (4.0–4.7)	0.023

* Continuous variables were compared using Student’s *t*-test or the Mann–Whitney U test, as appropriate. Chi-square or Fisher’s exact probability tests were performed for categorical variables. We determined the optimal cutoff points and significance level to be 5%. Abbreviations: M, male; F, female; APACHE II, Acute Physiology and Chronic Health Evaluation II; SOFA, Sequential Organ Failure Assessment; ISS, Injury Severity Score; CONUT score, Controlling Nutritional Status score; BMI, body mass index. High CONUT score group: CONUT score ≥ 5. Low CONUT score group: CONUT score ≤ 4.

**Table 3 diagnostics-14-01307-t003:** Independent predictor of a CONUT score of 5 or higher on day 7.

Explanatory Variable	Odds Ratio	95% CI	*p*-Value *
SOFA	-		
Operation	-		
Age	1.094	1.027–1.165	0.005
ISS	1.158	1.034–1.296	0.011

* Predictors of nutritional disorders were analyzed using multiple logistic regression with forced entry methods. Clinical factors considered to be related to nutritional disorders were used as explanatory variables. All variables with *p*-values less than 0.05 in bivariate models were analyzed using multivariate models (multiple logistic regression analysis). CI, confidence interval; SOFA, Sequential Organ Failure Assessment; ISS, Injury Severity Score.

## Data Availability

Data supporting the findings of this study are available from the corresponding author, J.Y., upon reasonable request.

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
