# Peer review of "Patients with Severe Trauma Having an Injury Severity Score of 24 and above Develop Nutritional Disorders"

_diagnostics, 2024, doi:10.3390/diagnostics14121307_

Round 1
Reviewer 1 Report
Comments and Suggestions for Authors
The paper deals with a rather specialized subject, the nutritional status and the correlation between ISS trauma score and teh CONUT score. The subject is well presented, and can be interesting for various subdisciplines, such as intensivists, nutrition scientists, traumatologists and neurosurgeons. For this reason, and although it deals with a specialized topic, I believe it deserves being published in Diagnostics.
Comments on the Quality of English LanguageMinor revisions required.
Author Response
Response to Reviewer 1 Comments
We are very pleased that you have reviewed our manuscript and appreciate your deep understanding of this topic. Thank you very much for your comments on the appropriateness of our submission. We have revised the manuscript in response to variable feedback on our manuscript after reviewing comments from the other reviewers. We look forward to hearing from you regarding our submission and to responding to any further questions and comments you may have.
Reviewer 2 Report
Comments and Suggestions for Authors
The message in this study is simple: trauma patients with a higher ISS score on admission will have more nutritional disturbances. The effects of malnutrition are many, on both increased in length of stay and mortality and need no elaboration.
The use of a simple prognostic score like CONUT is good as SOFA scoring was also developed due to the complexity of APACHE.
Thgere are a few queries:
1. What was the CONUT score of the subset of abdominal trauma cases (n = 5). Also what was the length of stay since the authors have declared that there was no mortality.
2. Regarding the energy intake, it is mentioned that it was over 7 days. What was the formula used for calculation of the requirement? BMI ranged from 20.3–25.6 while the range of energy intake was 4764.0–7321.0
3. The total number of surgeries as 29 (59%) is not understood. Are the authors stating that ~41% of cases were managed conservatively? The energy requirements of operative cases would be different from the energy requirement of non-operative cases. This should be highlighted. Also, some cases may have had more than one surgery, and the duration of surgery may have been different. This all should be highlighted as a separate subset of surgically managed cases of trauma.
4. The ISS range of 9.0-25.5 may also be divided with inter-range CONUT score for more clarification.
5. Concept of normal and light category is not understood.
6. References are not as per Vancouver style E.g., Waitzberg, D.L.; Goiburu. M.E.; Goiburu, M.J.; Bianco, H.; Díaz, J.R.; Alderete, F.; Palacios, M.C.; Cabral, V.; Escobar, D.; López, R.
Author Response
Response to Reviewer 2 Comments
We are very pleased that you have reviewed our manuscript and appreciate your deep understanding of this topic. We have revised the manuscript in response to your feedback.
We appreciate the time and effort you and the reviewers have dedicated to providing this valuable feedback on our manuscript, and we are grateful for the insightful comments. We have incorporated changes to reflect most of the reviewers’ suggestions and have highlighted them in the revised manuscript.
Below is a point-by-point response to the reviewers’ comments and concerns. We look forward to hearing from you regarding our submission and responding to any further questions and comments you may have.
